# Vaccine Hesitancy among Immigrants: A Narrative Review of Challenges, Opportunities, and Lessons Learned

**DOI:** 10.3390/vaccines12050445

**Published:** 2024-04-23

**Authors:** Jason Wong, Crystal Lao, Giancarlo Dino, Roujina Donyaei, Rachel Lui, Jennie Huynh

**Affiliations:** College of Pharmacy, Western University of Health Sciences, 309 E. Second St., Pomona, CA 91766, USA; crystal.lao@westernu.edu (C.L.); giancarlo.dino@westernu.edu (G.D.); roujina.donyaei@westernu.edu (R.D.); rachel.lui@westernu.edu (R.L.); jennie.huynh@westernu.edu (J.H.)

**Keywords:** immigrant, vaccines, vaccine hesitancy, immunization, barrier, COVID-19

## Abstract

(1) Background: Vaccination reluctance is a major worldwide public health concern as it poses threats of disease outbreaks and strains on healthcare systems. While some studies have examined vaccine uptake within specific countries, few provide an overview of the barriers and trends among migrant groups. To fill this knowledge gap, this narrative review analyzes immunization patterns and vaccine hesitancy among immigrant populations. (2) Methods: Four researchers independently evaluated the quality and bias risk of the 18 identified articles using validated critical appraisal tools. (3) Results: Most studies focused on vaccine hesitancy among migrants in the United States and Canada, with a higher COVID-19 vaccine reluctance than native-born residents. Contributing factors to this hesitancy include demographics, cultural views, obstacles to healthcare access, financial hardship, and distrust in health policies. Additionally, immigrants in North America and Europe face unfair vaccine challenges due to misinformation, safety concerns, personal perspectives, language barriers, immigration status, and restricted healthcare access. (4) Conclusions: Tailored vaccine education programs and outreach campaigns sensitive to immigrants’ diversity should be developed to address this issue. It is also important to investigate community-specific obstacles and assess the long-term sustainability of current efforts to promote vaccination among marginalized migrant groups. Further research into global immunization disparities among immigrant populations is crucial.

## 1. Introduction

Infectious diseases, also known as communicable diseases, can spread from person to person through direct or indirect contact. Vaccines are crucial in preventing the spread of diseases, reducing hospitalization costs, and even eradicating deadly infectious diseases. The World Health Organization (WHO) estimates that vaccines prevent around 2–3 million deaths worldwide yearly [1]. To protect vulnerable populations from serious complications of infection, the WHO has established a goal of achieving 90% international coverage of essential childhood vaccines [2]. However, global vaccine hesitancy has been rising recently, particularly due to the Coronavirus Disease 2019 (COVID-19) pandemic’s disruption of vaccination campaigns and routine childhood immunization services [3]. Vaccine hesitancy refers to delaying or refusing vaccination despite available immunization services [4]. It poses various challenges, including risks of outbreaks, re-emergence of diseases, public misconceptions and mistrust, health disparities in marginalized populations, and increased economic strain on global healthcare systems. Recent measles outbreaks, for example, highlight the urgent need to address vaccine hesitancy and under-immunization to reduce the risk of further outbreaks [5]. The advisory group behind the WHO’s global immunization practices and guidelines, known as the Strategic Advisory Group of Experts on Immunization (SAGE), has developed the 3C’s model of vaccine hesitancy, identifying the key determinants of vaccine hesitancy as confidence, complacency, and convenience [6].

The focus of this narrative review is immigrant populations, defined as people who consciously decide to leave the country of their birth to live in another country, usually permanently [7,8,9]. Further, some immigrants have undergone the entry country’s vetting process and unauthorized immigrants, and an individual’s status may change multiple times due to personal circumstances and changing policies. However, other groups of people migrate to another country for reasons outside of their control or for temporary durations, such as escaping violence, seeking employment opportunities, pursuing education, and reuniting families. Examples of these other groups include migrants, refugees, and asylum seekers. According to the WHO, refugees and migrants are some of the most vulnerable populations, facing xenophobia and discrimination as well as inadequate housing, working conditions, and access to healthcare services [10]. Migration may help people escape poor conditions but also expose them to barriers that diminish their health. Further, the migration process can be incredibly stressful, requiring a complete restructuring of daily life and having significant social, economic, and health consequences [11].

Vaccinating immigrants is an integral part of the global effort to promote vaccination, but not much research has been conducted on this population. The WHO has implemented the Immunization Agenda 2030, a global strategy to maximize the impact of vaccines and save 50 million lives over the next decade [2]. However, some studies have shown disproportionate mortality rates and lower vaccine uptake among immigrant populations [12]. According to Guttmann et al., 2020 [8], nearly half (43.5%) of total COVID-19 cases in Canada involve immigrants although their immigrant population only makes up a quarter (25%) of the country. Additionally, a more significant proportion of positive COVID-19 testing was recorded among the immigrant population, which revealed a disproportionate COVID-19 infection rate. Similarly, the immigrant population in Sweden, which only makes up 19% of the total population, was responsible for nearly a third (32%) of all positive COVID-19 cases as reported by The Public Health Agency of Sweden (Folkhälsomyndigheten) [13]. In Italy, Fabiani et al., 2021 [14], found that hospitalization and mortality rates were markedly greater among immigrant populations, particularly those from countries that measure lower on the Human Development Index. This overrepresentation of the immigrant population in COVID-19-related infection rates is likely due to factors such as religious beliefs, language barriers, mistrust or miscommunications, lack of access to healthcare and resources, transportation barriers, socioeconomic status, and cultural practices. Therefore, these factors must be understood to improve immigrant populations’ acceptance of vaccines and protect more people from communicable diseases.

Although studies on immigrant populations in the United States (US) and other individual countries exist, there is no global meta-analysis on this subject. This narrative review aims to identify and analyze current vaccination trends and vaccine hesitancy, barriers and challenges, lessons learned, opportunities to improve acceptance, and prospects globally among immigrant populations. By summarizing the current literature on vaccine hesitancy in immigrant populations, we aim to provide information to help promote vaccine uptake in this population.

## 2. Materials and Methods

Two co-authors independently conducted a thorough article search of electronic databases, using the Cochrane Library, Embase, PubMed, and Web of Science, from March 2019 to March 2024. The search strategy included the keywords “immigrant” and “vaccine hesitancy” combined with the Boolean search operators. In each database search, all locations and languages were included. The abstracts of the resulting articles were carefully reviewed. All articles that discussed vaccine hesitancy in immigrant populations and published within the last five years were included. Articles were excluded if they were “protocols” of studies, conference abstracts, case reports, articles not pertaining to research, and other miscellaneous non-narrative studies.

The articles included in this review underwent a rigorous appraisal process to ensure their quality and minimize potential biases. Four researchers independently evaluated each article’s quality and risk of bias. The following validated appraisal tools were used to assess for risk of bias: the AXIS tool was used for cross-sectional studies [15], the Newcastle–Ottawa Scale (NOS) for cohort and case–control studies [16], the Critical Appraisal Skills Programme (CASP) checklist for qualitative studies [17], the updated Assessment of Multiple Systematic Reviews (AMSTAR-2) scale for systematic reviews and meta-analyses [18], and the Scale for Assessment of Narrative Review Articles (SANRA) for narrative reviews [19]. There were no ethical concerns with using these analysis tools.

The two researchers, working independently, arrived at similar results during their literature searches in March of 2024. No filters were placed on language or location. Still, most results were English except for one Norwegian article that was not included because an English translation was unavailable. Although no specific filters were used for location, the majority of the locations in the research were mainly within Europe (i.e., Germany, France, and Sweden) and North America (i.e., Canada and the USA). Moreover, articles were filtered only to include publications from the last five years to obtain contemporary and up-to-date articles. Using the search terms “immigrant” AND “vaccine hesitancy,” the Cochrane Library produced 1 result, while Embase yielded 64, PubMed had 32, and Web of Science had 37. After reviewing the 134 articles for duplicates, 86 articles were identified. Of those, 18 met the inclusion criteria (refer to Figure 1). The majority of the articles (*n* = 11) focused on immigrant populations in the United States, with a smaller number examining populations in Canada (*n* = 4) and Europe (*n* = 3). The articles included a range of study types, including cross-sectional studies (*n* = 10), qualitative studies (*n* = 2), mixed methods studies (*n* = 2), systematic reviews (*n* = 2), narrative reviews (*n* = 1), and cohort studies (*n* = 1).

After critically appraising the included articles, researchers analyzed them for common themes regarding vaccine hesitancy in immigrant populations. Additionally, some of the articles were excluded from the analysis because they were not specific to the vaccine hesitancy of immigrants. Although they discussed non-immigrant groups, such as migrants or ethnic groups, similar factors might also provide insight into the immigrant experience. As such, these articles were discussed in the Discussion section, along with articles that were searched explicitly to add perspectives from immigrants from parts of the world outside of Europe and North America.

The authors would also like to acknowledge that the articles included for analysis in this work do not fully capture the immigrant experience globally, particularly in Asia, Africa, Australia, and South America. The exclusion of research based on these continents was not intentional, and the authors understand that there may be various reasons for this outcome. For instance, the research may not be available in an English translation, may not have been listed in the databases searched, or may not have included the keywords for the researchers’ literature search. Although the critically appraised articles primarily focused on Europe and North America, the authors tried to cover research from other parts of the world in the Discussion section.

## 3. Results

### 3.1. Vaccination Rates

When vaccine hesitancy is on the rise, it logically follows that vaccination rates will be affected. For instance, after working with Turkish-speaking family physicians to survey 420 individuals who spoke Turkish or German in Munich, Germany, Aktürk et al., 2021 [20], found that 348 (82.9%) had migratory backgrounds, with only 16 (4.6%) reporting to have already been vaccinated against COVID-19. In comparison, 7 (9.7%) non-immigrant Germans had been vaccinated (Fisher’s exact test = 3.027, *p* = 0.091). Additionally, only 145 (42.3%) of those with migratory backgrounds considered getting vaccinated, while 52 (76.5%) non-immigrant Germans expressed a willingness to do so (chi-squared = 6.818, *p* < 0.001).

In contrast, after surveying 296 adults with HIV in Seine-Saint-Denis, France, in 2021, Penot et al., 2023 [21], found that there was no significant difference in COVID-19 vaccine acceptance rates between immigrants (*n* = 197) and non-immigrants (*n* = 99) (66.3% vs. 76.3%). Additionally, Wanigaratne et al., 2023 [22], after analyzing a staggering 11,844,221 subjects using a Canadian health administrative database, presented differences in immigrants versus non-immigrants in Canada receiving two or three total COVID-19 vaccines. Overall, the study results showed that the vaccination rate in immigrants compared to non-immigrants was roughly the same for those receiving a two-dose series but much lower for those receiving a three-dose series.

Although Svallfors et al., 2023 [12], did not include a non-immigrant comparison group in their study, they had access to a vast pool of data from the Migrant World Values Survey from 2021. They examined COVID-19 vaccine hesitancy with various sociodemographic characteristics, values, and perceptions. Of the 2612 respondents who were first-generation immigrants in Sweden, 2011 (77%) reported their intention to get vaccinated or that they had already been vaccinated.

### 3.2. Barriers to Vaccination

The research findings in Europe and North America revealed that vaccine hesitancy is a complex issue influenced by various factors, such as misinformation, mistrust, socio-religious, and cultural reasons. These factors have a significant impact on people’s attitudes towards vaccines. For more detailed information on the results of each study, please refer to Table 1 and Table 2.

Aktürk et al., 2021 [20], shed light on the impact of misinformation and medical literacy on COVID-19 knowledge, attitudes towards vaccination, and behaviors related to the virus. Their survey comprised 25 items for knowledge, 7 for attitudes, and 7 for behaviors. The mean scores for knowledge, attitudes, and behaviors were 21.5 ± 3.2 (max = 25), 3.7 ± 0.8 (max = 5), and 4.0 ± 0.5 (max = 5), respectively. Although the scores were relatively high, 44 (12.6%) of those with migratory backgrounds agreed or somewhat agreed that COVID-19 does not exist. Additionally, 106 (30.6%) agreed or somewhat agreed that COVID-19 was created to control the world, compared to only 4 (5.6%) non-immigrants (chi-squared = 33.020, *p* < 0.001). The analysis revealed that migratory background (OR = 3.1), attitude scores (OR = 2.9), and sex (OR = 2.2) were the most significant variables affecting vaccination intention. The study also found that lack of access to healthcare and knowledge about vaccine ingredients and side effects contributed to vaccine hesitancy rather than outright opposition. Painter et al., 2019 [16], identified similar barriers after studying Latin American immigrant mothers with no insurance living in the US and examining their daughters’ uptake of tetanus, diphtheria, and acellular pertussis (Tdap); meningococcal conjugate (MenACWY); and human papillomavirus (HPV) vaccines. Despite the mothers’ overall acceptance of vaccines, a significant barrier the study identified was a lack of and even mixed knowledge surrounding vaccines.

Of the 2612 respondents surveyed by Svallfors et al., 2023 [12], 601 (23%) expressed some vaccine hesitancy. Although most respondents reported the belief that vaccination was necessary for their protection and that of others, some did not share such beliefs or were uncertain, and they were found to be at higher risk of vaccine hesitancy. Additionally, having a lower education level and low or uncertain trust in Swedish authorities were associated with a higher risk of unwillingness to get vaccinated. One’s home country was also found to influence vaccination intent, as those from Eastern Europe were more likely to refuse vaccination. However, the context of arrival was also important. For example, a significant portion of immigrants in Sweden in 2015 were from Afghanistan and Syria, and those who arrived in 2015 had a lower risk of being completely opposed or uncertain about vaccination.

Moreover, Penot et al., 2023 [21], discovered that administrative obstacles affected a smaller proportion of French-born individuals (9%) than immigrants (26.3%). Furthermore, financial insecurity and hunger were more frequently reported among immigrants (21.8% and 6.6%, respectively) than non-immigrants (7.1% and 3%, respectively). Despite no significant difference in overall COVID-19 vaccine acceptance rates between the two groups, non-immigrants were more likely to accept vaccines (56.7% vs. 32.1%) spontaneously. In contrast, immigrants were more likely to wait for their doctor’s recommendation (34.2% vs. 19.6%). Likewise, McFadden et al., 2022 [23], also identified socioeconomic barriers in the undocumented Hispanic immigrant communities in the US, such as fear of deportation, cost, limited access to information, side effects preventing them from working, language barriers, and concerns about DNA alteration.

In addition to mistrust and misinformation, fears and concerns about vaccine safety and efficacy have consistently impacted vaccine uptake. For instance, from a survey of 388 first- and second-generation Black immigrants across the US in 2021, Ogunbajo and Ojikutu, 2022 [24], identified fear of side effects and safety, mistrust in the government and the scientific community, and the rapid vaccine development as common themes of COVID-19 vaccine hesitancy. The study also found that immigrants who had not pursued higher education were less likely to be vaccinated. After studying Bhutanese, South Sudanese, Somali, Burmese, and Afghan refugee communities in the US, Zhang et al., 2021 [25], found that some individuals (*n* = 129 out of 435) were reluctant to receive the COVID-19 vaccine due to side effects, vaccine effectiveness, and fear of needles. Similarly, after surveying primarily Latino/immigrant participants in South Florida between 2020 and 2021, Kirchoff et al., 2023 [26], found that concerns around safety, side effects, and vaccine effectiveness, as well as negative information about the COVID-19 vaccine, largely drove vaccine hesitancy. In comparison, Sharp et al., 2024 [27], discovered that negative viewpoints towards the COVID-19 vaccine in Black, Asian, and Arab/Palestinian immigrant communities in Chicago, Illinois, were primarily due to skepticism surrounding the rapid development of the vaccine, lack of trust in the government, and a lack of autonomy.

Although some drivers of vaccine hesitancy can be clearly identified, people’s thoughts and actions tend to be multifaceted and require deeper examination. Case in point, Valero-Martínez et al., 2023 [28], found that older immigrant adults had mixed views about the COVID-19 vaccine, with reasons for not getting vaccinated including misinformation, lack of trust in the healthcare system and government, health-related concerns and underlying conditions, and systemic barriers. Sudhinaraset, Nwankwo, and Choi, 2022 [29], investigated the fear of immigration enforcement in relation to vaccine acceptance in California. Their study found that females and Latinx undocumented immigrants were more likely to receive the COVID-19 vaccine than their male and Asian counterparts. Additionally, Lin, 2022 [30], found that, of the 3522 people surveyed in Canada, after accounting for all covariates, the immigrant population was more apprehensive about receiving a COVID-19 vaccine (aOR 1.99, 95% CI 1.57–2.52) compared to native-born Canadians. These reservations included immigrants having a low perceived susceptibility (aOR 2.44, 95% CI 1.89–3.15) and a fear of being targeted/social norms (aOR 2.24, 95% CI 1.81–2.78). With this said, other reasons vaccine-hesitant immigrants (*n* = 596) refuse to receive a vaccine include safety (71.3%) and side effects (66.4%) of the vaccine, as well as misinformation/lack of trust (12.5%) in vaccines.

Aside from people’s beliefs regarding vaccines, there are tangible obstacles that immigrants must traverse in a new country, such as learning to communicate in another language, establishing new relationships, and navigating unfamiliar territory. Interestingly, a mixed-methods analysis of 57 studies by Daniels et al., 2022 [31], also identified knowledge gaps, lack of access to medical care, and cultural barriers as the primary factors contributing to vaccine hesitancy in refugees, immigrants, and migrants in the US. In Canada, there was a push to transition healthcare services to telehealth. Bajgain et al., 2022 [32], found that immigrants were more likely to not have a good telehealth experience compared to non-immigrant counterparts due to language barriers. Further, Malone et al., 2022 [33], presented a culturally sensitive approach to increasing vaccine uptake in Clarkston, Georgia, a city with a diverse population. The study highlighted the main factors contributing to vaccine hesitancy: mistrust and limited access to vaccine clinics resulting from location and language barriers.

Additionally, Stratoberdha et al., 2022 [34], investigated 34 studies about vaccine hesitancy and barriers to vaccination in Canada, with two studies specific to recent immigrants. The qualitative systematic review found that barriers to vaccination amongst immigrant populations include lack of access to vaccination, vaccine awareness (i.e., misinformation), and lack of vaccine information (i.e., lack of knowledge). These findings were primarily extracted from Zibrik et al., 2018 [35], which focused on HBV vaccination. The other article by McComb et al., 2018 [36], demonstrated that further barriers to vaccination include perceptions of resilience to HPV and a lack of vaccine information. Common themes of misinformation, knowledge gaps, and other health inequities (i.e., lack of access) are significant reasons for immigrant communities being more likely to hesitate when deciding to get vaccinated.

**Table 1 vaccines-12-00445-t001:** Overview of the included vaccine hesitancy studies.

Location	Study	Study Design	Risk of Bias (Appraisal Tool)	Study Time Period	Vaccine	Population Examined	*n*
Europe	Aktürk et al., 2021 [20]	Cross-sectional	Low-Moderate (AXIS)	February 2021	COVID-19 ^1^	Turkish- and German-speaking people with migratory background in Munich	420
	Svallfors et al., 2023 [12]	Cross-sectional	Low-Moderate (AXIS)	April to May 2021 and August to September 2021	COVID-19 ^1^	First-generation immigrants (age 16+ years) in Sweden	1809
	Penot et al., 2023 [21]	Cross-sectional	Low (AXIS)	5 January to 1 June 2021	COVID-19 ^1^	Immigrant adults with HIV living in Seine-Saint-Denis France	296
North America	Lin 2022 [30]	Cross-sectional	Low-Moderate (AXIS)	15–21 June 2020	COVID-19 ^1^	Im/migrants age 25+ in Canada	3522
	Bajgain et al., 2022 [32]	Cross-sectional	Moderate (AXISs)	October 2020	COVID-19 ^1^	16 years and up in Alberta, Canada (born in vs. outside of Canada)	10,175
	Stratoberdha et al., 2022 [34]	Qualitative systematic review	Moderate-High (AMSTAR 2)	1946 to January 2021 (MEDLINE) and 1974 to January 2021 (EMBASE)	HBV ^2^/HPV ^3^	Canada	
	Wanigaratne et al., 2023 [22]	Retrospective population-based cohort study	Low-Moderate (NOS)	13 September 2021 and 13 March 2022	COVD-19 ^1^	Canada	11,844,221
	Painter et al., 2019 [37]	Qualitative study	Moderate (CASP)	March to September 2016	Tdap ^4^, MenACWY ^5^, HPV ^3^	Latin American immigrant mothers living in Virginia, US	30
	Zhang et al., 2021 [25]	Cross-sectional	Moderate (AXIS)	December 2020 to January 2021	COVID-19 ^1^	Afghan, Bhutanese, Somali, South Sudanese, and Burmese refugee committees in the US	435
	McFadden et al., 2022 [23]	Narrative review	Fair/Poor Quality (SANRA)	September 2020 to April 2021	COVID-19 ^1^	Undocumented Hispanic immigrant community US	
	Sudhinaraset, Nwankwo, and Choi 2022 [29]	Cross-sectional	Low-Moderate (AXIS)	September 2020 to February 2021	COVID-19 ^1^	Undocumented immigrants in California, USA	326
	Malone et al., 2022 [33]	Case control	Moderate-High (NOS)	6 January to 28 May 2021	COVID-19 ^1^	Refugees from Bosnia, Kosovo, Liberia, Congo, Burundi, Sudan, Somalia, Ethiopia, Eritrea, Iraq, Syria, Bhutan, Burma, Afghanistan, and Pakistan living in Clarkston, Georgia, USA	3127
	Ogunbajo and Ojikutu 2022 [24]	Cross-sectional	High (AXIS)	January to February 2021	COVID-19 ^1^	1st- and 2nd-generation Black immigrants in the US	388
	Allen et al., 2022 [38]	Cross-sectional	Low-Moderate (AXIS)	July to August 2020	COVID-19 ^1^	Brazilian immigrant women ages 18+ in the US	353
	Daniels et al., 2022 [31]	Systematic review & Meta analysis	Moderate (AMSTAR 2)	April 2012 to May 2022		Foreign-born individuals ages 18+ that resettles within the US	
	Kirchoff et al., 2023 [26]	Cross-sectional	Low-Moderate (AXIS)	October 2020 to February 2021	COVID-19 ^1^	Adult Latinx immigrants in South Florida, USA	191
	Valero-Martínez et al., 2023 [28]	Mixed methods study	High (MMAT)	2021	COVID-19 ^1^	ethnically/racially diverse adults born outside of the US	100
	Sharp et al., 2024 [27]	Mixed methods study	Low (MMAT)	February to August 2022	COVID-19 ^1^	Immigrant communities in the Chicago metropolitan area	402

^1^ Coronavirus Disease 2019 (COVID-19). ^2^ Hepatitis B Virus (HBV). ^3^ Human Papillomavirus (HPV). ^4^ Tetanus, diphtheria, and acellular pertussis (Tdap). ^5^ Meningococcal conjugate (MenACWY).

**Table 2 vaccines-12-00445-t002:** Recommended approaches for addressing vaccine hesitancy (vaccine hesitancy): insights from the literature.

Study	Rates of Vaccine Hesitancy	Conclusions	Contributing Factors to Vaccine Hesitancy							Lessons Learned
			Mis- Information	Lack of Knowledge	Safety Concern	Personal Beliefs	Language Barriers	Immigrant Status	Access to Vaccine	
Aktürk et al., 2021 [20]	A total of 90.08% immigrant Germans surveyed did not consider becoming vaccinated	Turkish-origin people in Germany face social disadvantages and declining health across generations. Members of ethnic minority groups and vulnerable migrant populations exhibit reduced acceptance of vaccines.	X	X					X	To address vaccination hesitancy, clear and individualized information campaigns are important. Engaging trainers from the community can enhance acceptance and address distrust, potentially stemming from discriminatory attacks against immigrants.
Svallfors et al., 2023 [12]	A total of 25% of first-generation immigrants in Sweden surveyed (ages 16+) had some degree of vaccine hesitancy	Trust in healthcare providers and government authorities is crucial. Providing targeted vaccine information to those who face barriers to care can help make informed decisions about vaccination relating to health risks.		X						Governmental and healthcare entities need to confront various social determinants that contribute to low vaccination rates and disparities in health outcomes for first-generation immigrants
Penot et al., 2023 [21]	A total of 67.9% of immigrant participants held spontaneous unacceptance	The ICOVIH survey found that the already disadvantaged immigrant PLWHIV became more vulnerable during the first lockdown. However, they were confident in their physician’s recommendations and caught up with vaccination rates like those of PLWHIV born in France.			X	X				Patients depend on doctors to implement preventative measures for the most vulnerable populations, especially during the emergence and re-emergence of viruses, and with the development of new mRNA vaccines. Healthcare providers are essential in giving information and support to people dealing with chronic illnesses and social challenges.
Lin 2022 [30]	Not mentioned in the study	The research has shown differences based on birthplace, income, and education levels regarding COVID-19-related health concerns and reasons for refusing vaccination, even within the context of universal health coverage in Canada.	X	X	X				X	When organizing COVID-19 vaccinations in countries with many immigrants, patients, regardless of their legal status, can easily get vaccines and healthcare. The government and medical institutions should ensure fair access, and they must be responsible for this. Given the emergence of new virus variants, it is especially important to focus on health fairness and address the pandemic’s unequal impact on underserved communities.
Bajgain et al., 2022 [32]	Not mentioned in the study	The research underscores the importance of introducing a cost-effective community-based intervention, placing a priority on enhanced mental health care for all. This strategy is crucial for promoting and improving access to healthcare services.				X	X		X	The rising interest in virtual care highlights the importance of investing in technology and incorporating primary care to provide effective virtual care services. Patient education on the virtual care process is vital.
Stratoberdha et al., 2022 [34]	Not mentioned in the study	The immigrants of the Canadian population can greatly benefit from improving healthcare access and increasing vaccination. The best way to increase these rates is by helping individuals who are hesitant about getting vaccinated to change their mindset and become more accepting of vaccines.		X					X	Within the healthcare system, pharmacists should be highlighted to address vaccine hesitancy in patients. Vaccine hesitancy is handled in the pharmacy through involvement of vaccines.
Wanigaratne et al., 2023 [22]	Not mentioned in the study	Migrants have demonstrated repeating themes within vaccination that includes access, affordability, awareness, acceptance, and activation. Repeating themes in vaccination also include feeling a lack of trust towards healthcare and the government.						X		To increase participation of the bivalent COVID-19 vaccine, the healthcare system must gain the trust of the migrant communities.
Painter et al., 2019 [37]	Not mentioned in the study	In the study, participants demonstrated a lack of hesitation for vaccination amongst their daughters. Although there was little to no vaccine hesitancy, there was a clear lack of knowledge regarding general vaccines and history of immunization.		X	X				X	Amongst the repeating themes in the barriers of increasing vaccination rates, this study recommended finding ways to increase accessibility and education for vaccines. This would encourage the uninsured Latin American families to obtain their vaccines.
Zhang et al., 2021 [25]	A total of 29.7% had no plans on receiving COVID-19 vaccines	In this study, the authors surveyed refugees and discovered the participants held high acceptance towards COVID-19 vaccines.		X	X					To continue a strong acceptance in vaccines, the government and healthcare system must work with refugees to ensure vaccines remain accessible.
McFadden et al., 2022 [23]	Not mentioned in the study	This study attempted to understand the reason behind vaccine hesitancy. In this study, participants explained the reasons for uncertainty and would prefer to hold off to see how other patients may react to the vaccines.			X			X	X	This study recommends increasing accessibility to education regarding how vaccines are developed and explaining the importance behind vaccines to these hesitant patients.
Sudhinaraset, Nwankwo, and Choi 2022 [29]	Not mentioned in the study	This research suggests that immigration enforcement negatively affects health behaviors.						X		Enforcement tactics like surveillance, profiling, and deportation might target undocumented men more, leading to mistrust and fear of public health efforts. Asian undocumented individuals may be less accepting due to the increased anti-Asian attacks during the pandemic, fueled by rhetoric and policies from US government officials.
Malone et al., 2022 [33]	Not mentioned in the study	Elderly patients are often found to have a lack of access to vaccines. The lack of accessibility could be due to reasons such as transportation.	X				X		X	By addressing these concerns, elderly patients have a higher chance of becoming vaccinated. These concerns can be addressed by creating vaccination sites in greater regions to ensure accessibility in all communities.
Ogunbajo and Ojikutu 2022 [24]	A total of 43% would not get the COVID-19 vaccine immediately or at all	Black immigrants in the US have demonstrated increased vaccine hesitancy and tend to deny vaccinations. Some factors that contribute to vaccine hesitancy include less education, females, and working in healthcare settings.	X	X	X					Interventions to address vaccine hesitancy must address cultural barriers that immigrant populations face. One example of intervention could be offering education opportunities in healthcare.
Allen et al., 2022 [38]	A total of 29.2% were unsure or would not get vaccinated	In this study, participants demonstrated a lack of vaccine hesitancy when holding high confidence in healthcare providers and accepting the gravity of the pandemic.	X			X		X		To address vaccine hesitancy in Brazilian immigrants, education and emphasis of the severity of the pandemic is crucial. Information can be shared within the communities through Media channels within the local cultural context.
Daniels et al., 2022 [31]	Not mentioned in the study	In this study, the participants demonstrated potential willingness to alter vaccine hesitancy with higher accessibility to resources in healthcare.		X		X	X		X	Supporting a policy to ensure federal funding for health services and following ACIP guidelines for recommended immunizations will boost vaccination rates for RIM.
Kirchoff et al., 2023 [26]	A total of 32.5%	In this study, participants displayed vaccine hesitancy through themes of mistrust in the healthcare system and the safety behind vaccinations	X		X	X				To decrease vaccine hesitancy, the government and healthcare system must promote unity to the immigrant communities.
Valero-Martínez et al., 2023 [28]	Not mentioned in the study	This study observed immigrant participants in the elderly community. Participants exhibited vaccine hesitancy through themes such as mistrust, safety concerns, and systemic barriers.	X						X	To tackle these barriers, the government and healthcare system is encouraged to release more information and education regarding vaccines. Rollout programs would also be effective in addressing safety concerns.
Sharp et al., 2024 [27]	A total of 24% unvaccinated 5% partially vaccinated	Positive attitudes towards the vaccine stemmed from personal or community protection. The younger generation held higher reluctancy in getting the vaccine. In the African immigrant population, patients did not trust the government or healthcare system, due to the history of mistreatment in research and healthcare systems.	X			X			X	Plans and focus groups should be set in place to target the younger generations to build trust regarding vaccine hesitancy.

## 4. Discussion

### 4.1. Challenges, Opportunities, and Lessons Learned

The most frequent barriers to vaccination discovered within Europe and North America were cultural, social, religious, and physical barriers. This was followed by limited awareness and knowledge gaps surrounding vaccines, as well as low perceived susceptibility. Contributing factors to vaccine hesitancy were misinformation, lack of knowledge, safety concerns, personal beliefs, language barriers, immigrant status, and vaccine access. Moreover, most of the studies presented interesting perspectives on vaccine hesitancy, including “social stigma”, “loss of autonomy”, “fatalism”, “immigration status”, “parental consent”, “old age”, and “lack of formal education” [12,20,21,23,24,27,29,31,32,33,37,39,40,41].

Case in point, Bajgain et al., 2022 [32], found that native-born Canadians had more stress and anxiety during the pandemic than non-native-born Canadians, possibly due to social stigmatization of expressing any feelings or emotions commonly found in immigrant communities. It is intuitive, especially for those personally coming from an immigrant background, to understand that many immigrant communities find expressing or discussing issues surrounding their mental health and well-being to be somewhat taboo. Hence, it can be inferred that mental health, immigrant status, and vaccine hesitancy are potentially connected. There is an increased need to enhance telehealth for immigrant populations in hopes of creating a space for people to freely discuss mental health issues without fear of judgment, which will ultimately give rise to conversations about vaccine uptake in this already hard-to-reach population. Similarly, Aktürk et al., 2021 [20], pointed out that the rapid development of COVID-19 vaccines, combined with historical racist attacks on immigrants, has created an atmosphere of mistrust. Indeed, McFadden et al., 2022 [23], and Sudhinaraset, Nwankwo, and Choi, 2022 [29], found that immigration status can also be a barrier to vaccine uptake, particularly for undocumented immigrants.

Furthermore, in Aktürk et al., 2021 [20], many immigrants believed that COVID-19 does not actually exist or that it was purposely created to control the world. Interestingly, this study also found that negative attitudes towards vaccination were shaped by factors other than a simple lack of knowledge since many participants scored relatively high on knowledge tests. However, Ogunbajo and Ojikutu, 2022 [24], found that an individual’s education level possibly influences vaccine uptake, which is all the more enlightening when considering that marginalized groups (i.e., immigrants) in the US historically have had difficulty receiving formal education.

Interestingly, Svallfors et al., 2023 [12], concluded that the lack of information and education significantly contributed to vaccine hesitancy but not opposition. The study found that people desired more knowledge about vaccines, preferably from a trusted peer or doctor. The findings are consistent with the observation that individuals with lower levels of education are more likely to be hesitant or unwilling to get vaccinated. Additionally, the study suggested that trust in authorities was positively associated with vaccination in immigrant populations. However, trusting relationships with authority figures should be based on reciprocating respect towards an individual’s independence and concern about their well-being. For instance, Sharp et al., 2024 [27], found that lack of autonomy drives vaccine hesitancy in immigrant communities. Individuals forced to obtain the COVID-19 vaccine from their employers or face termination or other consequences may be less likely to get vaccinated. Furthermore, Daniels et al., 2022 [31], insinuated that fatalism, the belief that an individual’s health is entirely out of control and the worst is expected to come, may contribute to vaccine hesitancy. Autonomy is crucial in encouraging vaccine uptake because patients should have their voices heard in decisions regarding their well-being and health.

In another study, Holz et al., 2022 [39], conducted an online survey of first-generation migrants and native Germans in March 2021 to examine the relationship between migratory background and COVID-19 vaccination intention. The researchers found mixed effects of migration background, with increased religiosity leading to increased fear of infection and vaccination and increased trust in authorities and health consciousness. On the other hand, acculturation, years since migration, and the level of German media consumption showed conflicting associations with vaccination intention. Holz et al., 2022 [39], found that trust, health consciousness, and vaccination intention increased with these factors. However, previous research has shown that lower levels of acculturation are associated with higher vaccination rates, possibly because the disease being vaccinated against is still prevalent in the country of origin. It is clear that the role of migration background and acculturation in vaccination intention is complex, and further research is needed to understand these relationships fully.

Penot et al., 2023 [21], found no significant difference in vaccine acceptance rates between immigrants and non-immigrants. This finding is in contrast to other studies that suggest immigrants are more hesitant to receive vaccines compared to non-immigrants. For instance, Bajos, Spire, and Silberzan, 2022 [42], conducted a survey in France with more than 100,000 participants in 2020 and found that over half of the respondents were unsure of their attitudes towards COVID-19 vaccines. Further analysis revealed that first-generation African/Asian immigrants were more reluctant to vaccinate due to France’s colonial history, pharmaceutical scandals, mistreatment, and discrimination that they might have faced when interacting with the public healthcare system. However, Penot et al., 2023 [21], found that having a good relationship with a doctor responsible for treating their chronic illness was critical in determining vaccine acceptance in their study population.

Kour et al., 2022 [40], conducted interviews to explore the opinions and suggestions of Pakistani, Somali, and Iraqi immigrants in Norway regarding COVID-19 vaccine hesitancy. Using the “3C’s model of vaccine hesitancy” as a theoretical framework, the researchers identified five main themes that could help reduce vaccine hesitancy among immigrants in Norway. These themes are (1) effective cultural communication, (2) vaccine advocacy through community engagement, (3) motivating factors, (4) collaborative efforts via government and healthcare, and (5) incentives for vaccination. Overall, effective communication is crucial to all strategies. Immigrants, like everyone else, need to receive accurate information in their respective languages from sources they trust. They need to understand why vaccinations are necessary to be motivated to get vaccinated. Additionally, an open dialogue is essential to dispel misinformation and combat skepticism, which may have a long-standing history in their home country. Furthermore, efforts to reduce vaccine hesitancy should also address physical barriers, such as diminished mobility and lack of transportation, which were demonstrated in the elderly population of Malone et al., 2022 [33]. Conducting vaccine clinics within adult daycares, retirement/assisted living homes/communities, or local neighborhood school gyms or auditoriums could be a solution.

The perspectives of immigrant communities can help us address challenges and barriers that prevent everyone from receiving a vaccine. These ideas also provide lessons that can be applied to local and government health policies. In essence, the solution to the barriers that immigrants face is to provide care in a culturally sensitive and equitable manner.

### 4.2. Barriers to Childhood Vaccinations

In the case of childhood vaccinations within studies in North America, parents must be informed appropriately. For instance, Painter et al., 2019 [37], found that parental consent can be a barrier to vaccines, particularly in adolescents. In this study, it was found that nearly all mothers were the primary decision-makers and that their children had no say at all in the decision to be vaccinated. This is all the more interesting since the study also found that when mothers lack knowledge about vaccines, their children are less likely to get vaccinated. Furthermore, Khodadadi et al., 2022 [41], found that limited knowledge/awareness regarding the vaccine and the mothers’ perceived risk of HPV in their daughters were drivers in preventing HPV vaccine uptake. Overall, it is important to highlight the influence of parental figures and their lack of knowledge of vaccines and perceived risk as potential barriers to vaccination within children.

### 4.3. Global Perspectives of Vaccine Hesitancy

As previously stated, the articles included for analysis focused on Europe and North America. However, it is essential to note that vaccine hesitancy and immigrant populations can be found worldwide. This information can provide valuable insights into the challenges and opportunities related to vaccine uptake. According to the Pew Research Center, there were 281 million international migrants in 2020, with the majority residing in Europe, Asia, and North America [43]. With such a mass migration of people worldwide, it is crucial to decrease the spread of communicable diseases by decreasing vaccine hesitancy, which was declared one of the top ten threats to global health in 2019 by the WHO [1]. Vaccines already prevent 2–3 million deaths per year, and improved global vaccination efforts could potentially prevent 1.5 million further deaths. In the previously described literature search, some articles were excluded because they were not specific enough to the vaccine hesitancy of immigrants. For example, they focused on migrants or people of certain ethnicities, such as African Americans. While these groups may not have precisely the same experience as immigrants, there are overlapping issues that can influence their acceptance of vaccines. Therefore, even though these articles were not included for analysis in the review, they are discussed in this section along with other articles specifically searched to bring insight into parts of the world outside of Europe and North America.

To investigate attitudes towards vaccines during the COVID-19 pandemic, Harapan et al., 2022 [44], surveyed people across ten countries in Asia (Bangladesh, India, Iran, and Pakistan), Africa (Egypt, Nigeria, Sudan, and Tunisia), and South America (Brazil and Chile) from February to May 2021. Of the 1832 respondents, 665 (36.2%) were classified as vaccine-hesitant based on beliefs about vaccine benefits, and 1422 (77.6%) respondents reported vaccine hesitancy based on perceptions of the riskiness of new vaccines. Both types of vaccine hesitancy were significantly higher in females, people who identified as Muslims, people living in rural areas, people who worked non-healthcare-related jobs, and people who had not received a flu vaccine in the past 12 months. It is unknown why there are gendered differences, but it might be linked to limited safety and efficacy data of COVID-19 vaccines in pregnancy. Lack of information can lead to misinformation, especially for people who do not have in-depth knowledge of healthcare professionals. Furthermore, Harapan et al., 2022 [44], posited that vaccine hesitancy was influenced by some political leaders who claimed that COVID-19 is a conspiracy against Muslim countries.

Another example of vaccine mistrust was explored by Yu, Lasco, and David, 2021 [45], wherein the Philippines launched a mass immunization campaign against Dengue fever using the three-dose Dengvaxia vaccine developed by Sanofi Pasteur in 2016. However, in 2017, after more than 800,000 schoolchildren had received at least one dose of the vaccine, Sanofi announced that Dengvaxia may be unsafe in specific populations, prompting fear and mistrust from the community, not only towards vaccines but also towards other government health programs and the government itself. Dissemination of correct information from trusted sources in the community is essential for improved vaccine uptake and overall health outcomes.

In contrast, West et al., 2021 [46], found that vaccine hesitancy was low overall (25%) in temporary foreign workers from Bangladesh (*n* = 360) from 2020 to 2021. However, there was significant variation in vaccine hesitancy among host countries. The lowest vaccine hesitancy (14%) was found in Singapore and Malaysia (*n* = 81), while temporary foreign workers in other host countries had between 3.4- and 5.4-times higher odds of being vaccine-hesitant. Moreover, the study found that undocumented temporary foreign workers (*n* = 41) were more vaccine-hesitant (41% vs. 22%, *p* = 0.009) compared to temporary foreign workers with valid visas (*n* = 218) and had higher odds of being vaccine-hesitant (aOR 3.50, 95% CI 1.372 to 8.942). Additionally, temporary foreign workers with high COVID-19 threat perception, less worry about COVID-19 vaccine side effects, and employment-based risk exposures were less likely to be vaccine-hesitant.

Akintunde et al., 2023 [47], surveyed 498 foreign migrants in mainland China in 2021, of which 37.1% were students, and a significant proportion had a university or postgraduate level of education (92%). The findings showed that 17% of the participants had previously refused vaccines, and 30.7% had not received the COVID-19 vaccine. The analysis revealed that people with a history of vaccine refusal and an average monthly income of $1701 to $3500 were less likely to get vaccinated. This finding suggests that financial security may play a role in vaccine uptake. Additionally, the research found that individuals who rated their health as good compared to others were more likely to receive vaccinations. Healthy individuals may be more proactive in maintaining their health. However, more research is required to explore vaccine uptake among unhealthy individuals.

Achangwa, Lee, and Lee, 2021 [48], surveyed 710 foreigners residing in South Korea in 2021, of which 29.2% reported hesitancy towards the COVID-19 vaccine. The primary reason behind this hesitancy was concern about side effects. Other reasons included concerns about vaccine safety, ineffective vaccines, and lack of confidence in government policies regarding COVID-19 vaccines. The study also found that doctors’ recommendations, past acceptance of vaccines, convenience, safety, and efficacy were positively associated with willingness to get the COVID-19 vaccine. The relatively quick development of the COVID-19 vaccine instilled doubts in the population, which can be addressed by education from trusted sources.

Trust is crucial in vulnerable and marginalized populations, such as refugees. Liddell et al., 2021 [49], surveyed a cohort within the Refugee Adjustment Study in Australia in June 2021. The study found that, out of 516 participants, 439 (88%) were unvaccinated. Of those unvaccinated, 123 (28.1%) hesitated to get vaccinated, and the most common barrier was the lack of clear information about the vaccine (*n* = 237, 54.5%). According to the study, mistrust in the COVID-19 vaccine (OR 0.87, 95% CI 0.57 to 1.33) and attitudes relating to a lack of control (i.e., being forced to get vaccinated) (OR 1.65, 95% CI 1.21 to 2.24) were associated with greater vaccine hesitancy. Additionally, trust barriers were linked to greater vaccine hesitancy at the second level (OR 2.00, 95% CI 1.39 to 2.90) and third level (OR 1.70, 95% CI 1.11 to 2.60) of logistic regression. This mistrust could be because of past experiences with prosecution, conflict, and inadequate protection from authorities in their home country. However, the majority of the participants still intended to get vaccinated. Establishing a strong relationship between the refugee community, healthcare workers, and other authority figures is critical to disseminating correct information and building trust.

Tankwanchi et al., 2021 [50], conducted a rapid review to identify the drivers of vaccine hesitancy, which were similar to the SAGE Group’s three C’s. Convenience-related structural drivers were found to be particularly apparent in Romanian and Roma Romanian populations in the UK, where under-immunization was primarily caused by language and literacy barriers, as well as financial cuts to services aimed at underserved communities. However, systemic factors are not the only determinant of vaccine uptake. Complacency-related drivers regarding HPV and influenza vaccinations were observed in parents. In particular, Somali immigrants in the US and the Netherlands perceived a low risk of HPV due to their Muslim faith, which precludes premarital sex. Limited knowledge of cervical cancer and social conservatism have hindered informed discussions of sexual and reproductive health, further hindering HPV vaccine uptake. Moreover, Polish immigrant families in the UK have shown significantly higher rates of declining flu vaccination in their children, primarily due to the continued influence of the anti-vaccine movement in Poland. Low confidence in vaccinations is yet another factor in vaccine hesitancy. The most notable example is the fears incited by the false report of MMR vaccines being linked to autism, which prompted anti-vaccine campaigns targeted towards Somali communities in Minnesota and a subsequent decrease in MMR vaccination in Somali children, from 92% in 2004 to 42% in 2016. In a more thorough review, Tankwanchi et al., 2022 [51], reached similar conclusions that individual-level behavioral variables and broader structural factors need to be addressed to improve vaccine uptake in migrant populations.

Deal et al., 2023 [52], found that migrants’ opinions on vaccinations were significantly influenced by their trust in the vaccine and the host country’s more comprehensive government and health systems. Some African migrants in the UK had concerns that vaccination campaigns in Africa contained hidden agendas of “the West”. Additionally, access to vaccine information depended on the migrant’s understanding of the language and media formatting. Campaigns in Australia, Canada, Japan, South Korea, and the USA were translated into different languages. Still, they were criticized for being delayed, containing a lack of details, and having a lack of diversity in available languages. Therefore, multiple formats should be used to extend the reach and provide alternative access routes. As previously mentioned, political and economic factors could also affect vaccination. Undocumented migrants may be hesitant towards vaccination due to fears of data collection, lack of eligibility, and immigration checks. However, Deal et al., 2023 [52], noted that some countries have allowed COVID-19 vaccination to be anonymous without links to immigration enforcement, such as Colombia, which enacted a 10-year protection status for Venezuelan migrants, allowing them to register for vaccination. When developing methods to overcome barriers to vaccination, indirect costs, including transportation and wages lost from taking time off work, should be considered in addition to the direct cost of the vaccine. For example, Deal et al., 2023 [52], mentioned programs that bring vaccines to migrants being successful, as seen in the more than 20,000 childhood vaccines delivered in a door-to-door program for refugees in Greece.

### 4.4. Applications in Pharmacy

Community pharmacists are among the most easily accessible healthcare providers and receive extensive vaccine training. Due to their expertise, they are in an excellent position to promote vaccine uptake, increase awareness, and educate the community. Pharmacists in immigrant communities play a vital role in eliminating language barriers and filling knowledge gaps to encourage vaccine uptake. Culturally sensitive care from pharmacists can help overcome vaccine hesitancy in immigrant populations and dispel vaccination misconceptions [53,54].

### 4.5. Limitations

The authors of this narrative review strived to conduct a rigorous and unbiased search and appraisal of the available literature regarding vaccine hesitancy in immigrant populations. However, the literature search did not include every available database so this review may have missing information, particularly articles published in non-English languages. Although no filter was placed on the geographic location of studies, the articles analyzed in this review primarily focused on immigrant populations in the United States, Canada, and Europe. As such, the conclusions drawn from this review may not be generalizable to the world at large, especially countries in Africa, Asia, Australia, and South America. Additionally, articles were filtered to include only those published within the last five years. Although this filter was intended to ensure that contemporary and up-to-date articles were examined, the authors acknowledge that most of the articles focused on COVID-19 vaccine hesitancy, despite a minority focusing on other immunizations. Furthermore, the review was specifically focused on immigrant populations. However, the authors recognize that different migratory patterns and social factors associated with ethnicity and race may also affect vaccine hesitancy. For example, people who are documented as opposed to undocumented may have different experiences with healthcare systems in a new country. Moreover, one generation may not be immigrants, but they may have descended from previous generations of immigrants, which may influence their experience in society and perception of vaccines.

The WHO and CDC have identified Social Determinants of Health (SDOH), which are “the conditions in which people are born, grow, work, live, and age, and the wider set of forces and systems shaping the conditions of daily life” [55,56]. Examples of SDOH include education, housing, (un)employment, early childhood development, social inclusion and non-discrimination, and access to affordable healthcare of adequate quality. Across many countries, it is commonly recognized that the lower one’s socioeconomic status, the worse one’s health [55]. Historically, socioeconomic status, race, and ethnicity have been consistently intertwined, with communities often being segregated based on these factors. Particularly, in the US, there have been significant gaps in income and education when comparing minorities to Whites, hindering upward mobility and access to healthcare [57]. Furthermore, in the US, minority populations have experienced higher rates of heart disease, obesity, diabetes, and asthma, as well as lower life expectancy compared to White counterparts. Even the COVID-19 pandemic disproportionately impacted minorities and resulted in higher rates of hospitalization and death [58]. Although the issues of racism and discrimination are significantly highlighted in the US, people from across the globe have experiences with this stigma, such as Indigenous peoples, people of Roma descent, and people of African descent. These inequities are made apparent in both structural and interpersonal racism, which can severely impact health outcomes when demonstrated in health systems and healthcare professionals [59]. Although many of the articles included in this narrative review discussed such factors concerning immigrants’ vaccine hesitancy, not all of them compared that data to non-immigrants. Both immigrants and non-immigrants may belong to other racial, ethnic, or religious groups, which may influence their health outcomes. As such, there might be a gap in understanding what differences affect these groups of people regarding vaccine hesitancy. Furthermore, it can be challenging to determine how much influence the societal norms of one’s home country may have on vaccine hesitancy after moving to another country.

## 5. Conclusions

The findings in our study suggest that immigrants within Europe and North America (Canada and the United States) are disproportionately dealt with vaccine-related inequities. Common factors contributing to vaccine hesitancy include misinformation, mistrust, lack of knowledge about vaccines, safety concerns, personally held beliefs (i.e., cultural/religious beliefs and social norms), language barriers, immigration status, and a lack of access to the vaccine. Addressing these factors is particularly important when communicating with parents about their children’s vaccinations. With this said, the primary method to promote vaccination within these marginalized populations is to provide vaccines and healthcare, in general, in a culturally appropriate and sensitive manner.

The findings of our review carry significant implications for global health agencies. It highlights the need to develop vaccine education programs and outreach campaigns catering to immigrant populations’ linguistic and cultural diversity. Such programs can help address the challenges and barriers preventing immigrants from accessing vaccines. Furthermore, this review suggests a need for future research on vaccine disparities among different immigrant communities, both nationally and globally. Researchers could explore cultural and socioeconomic factors contributing to vaccine hesitancy in immigrant populations. Finally, there is a need to appraise the long-term sustainability of current vaccination programs for immigrant populations. This can help health agencies ensure that their vaccination programs are effective in the long run and continue to meet the needs of immigrant populations.

Our narrative review of vaccine hesitancy within the immigrant communities in Europe and North America highlights the pitfalls of vaccine dissemination within this population and the importance of promoting healthcare in a culturally sensitive manner.

## Figures and Tables

**Figure 1 vaccines-12-00445-f001:**
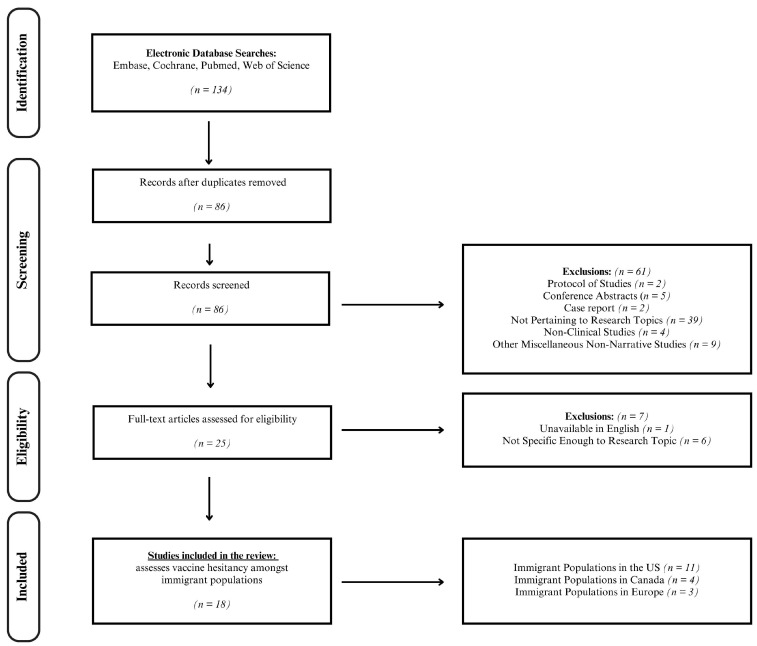
A flow chart of the article selection process.

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
