# Peer review of "Vaccine Hesitancy among Immigrants: A Narrative Review of Challenges, Opportunities, and Lessons Learned"

_vaccines, 2024, doi:10.3390/vaccines12050445_

Round 1
Reviewer 1 Report
Comments and Suggestions for Authors
The manuscript by Wong et al. reviews a number of problems immigrants face that increases their vaccine hesitancy. For countries, like the US, which has a considerable immigrant population, this is an important issue. So, the topic is important.
The method is a bit odd. A meta-analysis or a systematic review requires such well-documented search. My problem is not that the authors documenting how they arrive at the papers to include in the review, but the lack of try to get other papers, especially in the light of what they write in section 4.3. First, have they tried to include papers cited in the found articles that point to articles not found or in another language? Some of the retained papers are (systematic) reviews themselves, and as such should have some more papers in them to consider. The keywords seem to be rather too specific (immigrant).
Section 4.3 lists studies that seems to be very relevant to this study. But they are not found in the tables. For example, how come that the Tankwanchi papers mentioned in the discussion are not included in the tables? Why did they not fit into the selected papers? Also, some of those papers come from outside of North America and Europe and would broaden the scope of the review.
I suggest to also consider cleaning the structure of the manuscript a bit. For me it felt random why some of the results of the found papers are mentioned in the Results and some in the Discussion.
Consider including information in Table 2 about the vaccine hesitancy of the non-immigrant population. That would put the number more into perspective. It would be interesting (but I do not feel it is a must) to mention the vaccine hesitancy of the sender countries. For example, if it is somewhat cultural, then we might find similarly low rates at the country of origin and among the immigrants. But if it is just about not understanding the local language, being mistrustful of local (strange) authorities, unavailability of information, then vaccine hesitancy should be lower among immigrants compared to both natives of the sender country and the receiving country.
Vaccines hesitancy is an important issue, especially when it relates to vaccination against deadly diseases. Most of the information presented here comes from hesitancy toward COVID-19 vaccination. I do not want to downgrade how important COVID-19 vaccination was in stopping the pandemic, and it would still be important to stem the new waves. It would still be important to know how people with immigrant background relate to childhood vaccination. The study by Painter et al. (2019) is important specifically and should be presented in more details. I’m from Central-Eastern Europe, and while the acceptance of childhood vaccination is high here (and we have one of the most comprehensive vaccination regime), hesitancy toward COVID-19 vaccines were higher compared to Western Europe.
In summary, with restructuring and more focus in how the small set of paper’s finding is presented this manuscript can be considered for publication in Vaccines.
Minor comments
Lines68-74: Please cite the articles after the protocol / tool mentioned a not lumped together. For example [8] after AXIS, [9] after NOS, and so on so forth.
Do not abbreviate vaccine hesitancy as VH. There is no character limit, and the article is easier to read without having to constantly unfold an abbreviation in our mind.
Furthermore, temporary foreign workers should not be abbreviated as TFW. It only appears five times and all of them in the same paragraph.
Line 129: “The research findings revealed that VH is a complex issue influenced by various factors.” This sentence is too general and does not say much. At least list some of the factors.
Line 256–257 “Immigrants may hide their feelings and emotions because of social norms, which could impact vaccine uptake.” This statement needs more elaboration. It is intuitive to me that in some culture mental problems are still not something people freely discuss. But vaccines are not treating mental problems nor are they supposed to induce ones (with the many times over debunked “causing autism” stuff). So, what is the connection between mental illnesses, immigrant status and vaccine hesitancy?
Table 1. Why does only some of the studies have citations (bracketed number)?
Citations are odd. It seems to be a mix of numbered ones (which is the format the journal uses) and author names and year cites. First the number should be after the mention of the authors and not at the end of the sentence. Second, it should be consistent to have author names and years when specifying articles. Sometimes the year is omitted but at other times it is there. Table 2 and 3 has many of these inconsistencies.
Author Response
Dear Reviewer,
Thank you for taking the time to review our work. We appreciate your feedback and have addressed your comments in the attached file. Our responses are in bold, following your feedback. Thank you for your help and attention.

Reviewer 2 Report
Comments and Suggestions for Authors
Vaccine hesitancy (VH) is not only a common problem among immigrants, it is similar among nonimmigrants too. Basically, there is no significant difference of vaccine acceptance among immigrants and nonimmigrants. But the reasons are different. In this review the authors demonstrated nicely the reasons of VH among immigrants specially in United States, Canada and Europe. They also discussed some of the barriers and how to overcome these barriers of VH among immigrants. It would be better if they discuss some of the challenges of VH among nonimmigrants and compare them with immigrants.
Author Response

(The authors gave the same response as above.)

Reviewer 3 Report
Comments and Suggestions for Authors
Abstract
The study objectives are to be reviewed and include only on Europe and North America. This is not a global review!!
Keywords:
Introduction
Instead of providing general context, the first section of the introduction should center on coverage and other pertinent information regarding the immigrant population.
Line 52-54, page 3. However, some studies have shown disproportionate mortality rates and lower vaccine uptake among immigrant populations [7].
Comments: the authors informed that some studies and what I I see only one study; could you add some more?
Materials and Methods
This part is too short and lacking key information.
The methodology should include the research period. The nations or regions covered by the research. The process of synthesizing data and determining who collected it.
Which analysis was implemented, and are there any ethical issues?
Describe how the data was analyzed for common themes. What are these themes?
Have included only English literatures ?
Results
Line 86-94, page 2
Comment: the information given to be part of the materials and methods
Line 92-95, page 2. You inform that the literature search were primarily focused on Europe and North America, the authors made an effort to cover research from other parts of the world in the Discussion section.
Comment: The authors need to revise the study objective based on the above and across the manuscript
Discussions
Comment: Start with key study findings
There a lot of repetition from the results, I suggest to summarized the key findings and discuss around it
Line 247-252, page 12
Comment: provide the references for the study narrated
Author Response

(The authors gave the same response as above.)

Round 2
Reviewer 1 Report
Comments and Suggestions for Authors
My comment and suggestions were taken into account and I have no further suggestions.
Author Response
Thank you for the second review and all the feedback you provided. We appreciate it.
Reviewer 3 Report
Comments and Suggestions for Authors
NIL
Author Response
Dear reviewer,
Thank you so much for your review and all the feedback you provided. We appreciate it.